# Mastering SAM Prompts: A Large-Scale Empirical Study in Segmentation Refinement for Scientific Imaging

**Stephen Price**                                                             *sprice@wpi.edu*
*Department of Computer Science, Worcester Polytechnic Institute*

**Danielle L. Cote**                                                          *dlcote@wpi.edu*
*Department of Mechanical and Materials Engineering, Worcester Polytechnic Institute*

**Elke A. Rundensteiner**                                                     *rundenst@wpi.edu*
*Department of Computer Science, Data Science & AI Program, Worcester Polytechnic Institute*

**Reviewed on OpenReview:** *https://openreview.net/forum?id=cWcTQMpqv6&noteId=SoSjGE2AL2*

## Abstract

Segment Anything Model (SAM) has emerged as a prevalent tool empowering advances in vision tasks from instance segmentation, panoptic segmentation, to interactive segmentation. Leveraging powerful zero-shot capabilities enabled by visual prompts such as masks placed on the image, SAM has been shown to significantly improve tasks. Yet, a poor prompt can worsen SAM performance, risking consequences such as misdiagnoses, autonomous driving failures, or manufacturing defects. However, recent studies on visual SAM prompting remain limited, cover only a small fraction of potential prompt configurations, adopt ad-hoc evaluation strategies, and come with limited or even no rigorous analysis of the statistical significance of prompt configurations. To address this gap, we undertake the first large-scale empirical study comprehensively evaluating the impact of SAM prompt configurations on segmentation refinement. This includes 2,688 prompt configurations, including points, boxes, and masks with diverse augmentations, on four initial segmentation models for a total of 10,752 evaluations. From these results, we draw statistically significant insights along with practical guidelines for prompt design on scientific images. In particular, we recommend including a bounding box, which raised AP@50-95 by 0.320 and advise against using a coarse mask, which lowers AP@50-95 by -0.133 across all four models on microscopy data sets. We showcase that our recommended prompt configuration enables SAM to outperform leading refinement methods on multiple scientific benchmark datasets.

## 1 Introduction

**The Promise of Segment Anything Model (SAM).** The recent introduction of Segment Anything Model (SAM) by Kirillov et al. (2023) has revolutionized the field and practice of computer vision, streamlining numerous tasks that had previously been significantly more challenging. SAM has enabled improvements in weakly supervised instance segmentation (Wei et al., 2024), high-resolution object segmentation (Ke et al., 2024), zero-shot segmenting (Yamagiwa et al., 2024), and 3D-object detection (Zhang et al., 2023b). Impressively, SAM leverages zero-shot prompting to achieve these advancements across a diverse range of tasks. User-specified visual prompts, such as points, boxes, or coarse masks, guide the model's outputs to the desired task without requiring additional training or prohibitively expensive modification to the massive model architecture. In this work, we leverage SAM for segmentation refinement of scientific and medical microscopy (Figure 1a), converting initial segmentation masks into prompts and producing high-quality refinements. We focus on refinement because it supports rapid creation of large numbers of input masks, enables automated prompt derivation per instance according to defined criteria, and allows high-fidelity quantitative evaluation against ground-truth IoU.

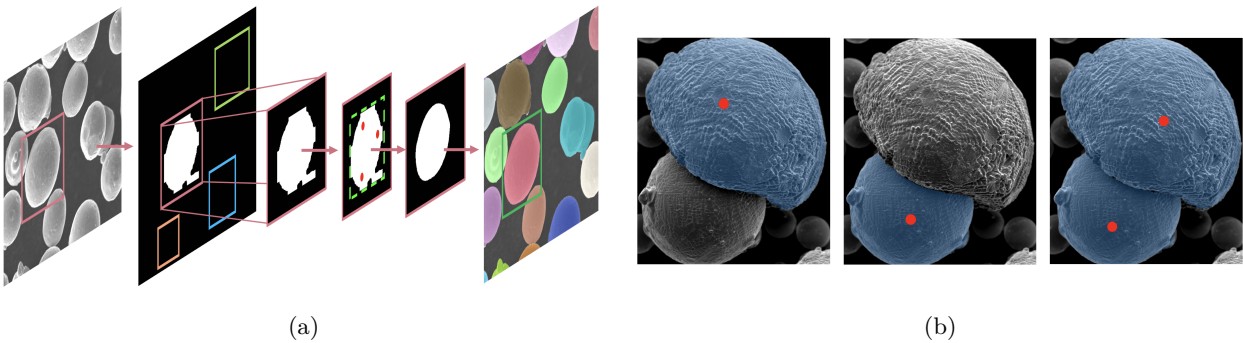

(a)                     (b)

Figure 1: **(a)** *Example segmentation refinement pipeline using SAM*, including making initial predictions, processing them into prompts for SAM, producing refined segmentations, and aggregating across all objects. **(b)** *Impact of example prompt configurations* on SAM segmentations, showcasing how variations can drastically alter final segmentations (red dots denote a prompt placement of points as visual prompts).

**Limitations of SAM.** Despite extensive work integrating (Wei et al., 2024), extending (Ke et al., 2024), or fine-tuning SAM (Wu et al., 2025), there is a lack of systematic evaluation of how prompt design affects its performance. While SAM has the potential for strong zero-shot capabilities, SAM's accuracy is highly dependent on the "quality" of its prompt. For example, a single point-prompt might cause SAM to under-segment the desired object, as shown in Figure 1b. By contrast, using multiple point-prompts can improve coverage of the object's boundaries but requires careful selection of point locations – but again too many or poorly placed points can inadvertently force SAM to include surrounding regions or merge adjacent objects (Wei et al., 2024). Additionally, SAM's performance is domain-dependent, often underperforming on low-contrast, fine-grained scientific and medical microscopy images essential for critical applications (Ma et al., 2024). In this work, we therefore concentrate on prompt design for this important class of scientific and medical applications that leverage microscopy images to derive practical guidelines.

**Research Questions.** Given the variability in image resolution, object morphology, and desired prompt attributes, the number of unique prompts is near-infinite. As a result, open questions remain regarding prompt design composition. In this work, we conduct a rigorous study of performance trends to guide prompt selection and answer the following research questions in microscopy settings: What is the relative importance of boxes, masks, and point prompts on segmentation quality? How do different potential types of visual augmentation strategies (e.g., point-placement strategies) impact SAM's results? How does combining different prompt types and their visual augmentations affect segmentation quality, in particular, what are the challenges and promises derived by the potential interaction (amplification, attenuation, or even cancellation out of the positive or negative impact) of one prompt type on others when combined into visual prompt configurations? Finally, with SAM's training set, SA-1B, primarily including natural-scene high-contrast benchmarks such as COCO or Cityscapes rather than the *low-contrast, small-scale, and irregular features* common in *scientific and medical images* (Li et al., 2025), if and how best can SAM's performance be effectively adapted to the unique challenges in these scientific and medical domains?

**State-of-the-Art and its Shortcomings.** Consequently, recent studies have found that SAM's out-of-the-box performance often falls short on segmentation tasks in these important scientific and medical application domains (Ma et al., 2024; Zhang et al., 2024c). With the growing demand for generalizable yet high-precision segmentation models in these important domains—including cancer detection (Kassis et al., 2024), high-throughput microscopy (Rusanovsky et al., 2022), histopathology (Sikaroudi et al., 2023), and surgical planning (Min et al., 2025), establishing effective visual prompting strategies to achieve high-quality segmentation for these settings is imperative. Without an effective prompting strategy, these works implemented alternative strategies, such as retraining SAM (Ma et al., 2024), relying on human-in-the-loop interactive segmentations (Shen et al., 2024), or both (Cheng et al., 2023b).

Recently, Cheng et al. (2023a), Mayladan et al. (2023), and Dai et al. (2023) have begun to explore SAM prompting across natural, medical, and remote-sensing data. However, these studies have been exceedingly

small, typically evaluating fewer than 25 prompt configurations. With so few combinations, these studies have not statistically validated the significance of each prompt type or of its potential types of visual augmentations. Lastly, these works typically explored a single prompt type (points, bounding boxes, and coarse masks) at a time, overlooking the problem of determining the complex interactions of prompt types and their augmentations when combined into visual prompt configurations.

**Our Approach: Large-Scale Analysis of Prompt Design Configuration.** To address this gap, we undertake a comprehensive study of visual prompt design configurations and their combined impact on the performance of SAM for segmentation refinement in scientific images. Our rigorous study covers 2,688 prompt designs on four alternate popular initial segmentation models, resulting in 10,752 evaluations. In particular, we first identify the effect of each core prompt type (points, masks, and boxes) and their unique augmentation strategies. Namely, for boxes, uniform scaling is employed to expand or contract the as-detected boundary while preserving spatial location. For masks, a similar approach is considered, but with additional requirements to ensure the preservation of the object's morphology. For points, we evaluate multiple strategies, including adjusting the number of points placed, tailoring the search space for placement, and modifying the placement algorithm. Next, we also thoroughly explore their prompt type interrelationships by creating composite prompt configurations that include (or exclude) each core prompt type, as well as their respective augmentations.

Throughout this evaluation, we discovered *significant prevalent trends* that serve as a foundation for deriving guidance for practitioners for prompt design. For example, prompts that contained bounding boxes improved performance by 0.320 AP@50-95 through a wide array of testing scenarios on scientific images. Alternatively, adding coarse masks reduced it by -0.133 AP@50-95, and point-based components could boost accuracy but were dependent on the number of points and their spatial placement. To validate the robustness and generality of these trends, we then extended our analysis of the top prompt's performance to determine its impact on other metrics, including individual IoU thresholds (AP@50, AP@75, AP@95), revealing both bulk and fine-grained improvements. We further explored the robustness and generalizability of our evaluation by applying our top-performing prompt configuration developed on one key data set as is, without further prompt refinement, to other benchmark scientific datasets not considered during the prompt design study. For this, we then compare SAM's zero-shot refinement performance using our identified recommended prompt configuration against the leading model-agnostic segmentation refiners, CascadePSP (Cheng et al., 2020) and SegRefiner (Wang et al., 2024). Overall, our study highlights that SAM, with our well-designed prompt, can achieve significant improvements in segmentation quality on scientific images, a domain where it has previously been found to struggle (Ma et al., 2024).

**Contributions.** In summary, this work offers the following contributions:

- The largest empirical study of SAM prompt design to date, covering 2,688 unique configurations and four initial popular segmentation models, totaling 10,752 evaluations. This study covers the first systematic analysis of how points, boxes, and masks interact when combined in a prompt configuration.
- Comprehensive evaluations demonstrating statistically significant trends for prompt design, leading to valuable guidelines for practitioners for the utilization of SAM in scientific and medical domains.
- An experimental validation of SAM's model-agnostic performance compared to state-of-the-art techniques, particularly designed to tackle the segmentation refinement problem across three scientific image datasets, four initial segmentation models, and four performance metrics.

## 2 Related Works

### 2.1 Segmentation Refinement Approaches

Segmentation refinement refers to the task of processing an initial model's output to yield an improved new mask (Tang et al., 2021). These methods fall into two categories: *model-dependent* and *model-agnostic solutions.* Model-dependent solutions, such as RefineMask (Zhang et al., 2021), PointRend (Kirillov et al., 2020), and Mask Scoring R-CNN (Huang et al., 2019), add additional layers to a model's architecture to produce higher resolution or more accurate segmentations. However, these approaches are tailored to a specific model architecture and, as a result, typically cannot be used elsewhere. Model-agnostic approaches,

such as CascadePSP (Cheng et al., 2020) or SegRefiner (Wang et al., 2024), instead post-process an initial model's segmentation. The latter eliminates the dependency on the specific architecture, offering improved versatility across domains. For this reason, we henceforth focus on the latter approach.

**Model-Agnostic Segmentation Refinement.** Prior to the introduction of SAM, model-agnostic segmentation refinement approaches were not based on prompting. One of the first refinement methods, CascadePSP (Cheng et al., 2020), applies a cascade of boundary-aware residual modules on a coarse mask, progressively enhancing local edge detail. SegRefiner (Wang et al., 2024) at NeurIPS 2023 was the first to integrate diffusion-based models into the refinement process, building on previous work using GANs (Le, 2020). Most recently, Meta's Segment Anything Model (SAM) (Kirillov et al., 2023), a foundational computer vision model, has emerged as a highly competitive segmentation refiner (Lin et al., 2025; Yu et al., 2024; Wei et al., 2024; Chen et al., 2024; Mayladan et al., 2023). When using SAM, the initial segmentations are used to derive visual objects that serve as prompts, such as bounding boxes, coarse masks, and points, to aid SAM's re-segmentation (See Figure 1a).

## 2.2 SAM and Visual Prompting

Previous works, while starting to analyze the effect of prompting on SAM performance, have been quite limited. In particular, prompt types are frequently analyzed in isolation. For example, Hu et al. (2023) briefly tested how increasing the number of points from 1, 5, 10, or 20 affected performance but did not specify how points were placed and ignored any other prompt types. Alternatively, Cheng et al. (2023a) performed a preliminary analysis of six bounding box prompts and three-point prompts. However, there was no variation in the point-prompting method. They neglected to evaluate when points and boxes are used together. Dai et al. (2023) explored four point-placement strategies but constrained their experiments to only two points. While they explored a few box prompts, similarly to prior work, they did not explore points and boxes integrated together into one unified prompt configuration. Lastly, Mayladan et al. (2023) explored two prompts containing a box and points – testing one and five points only, plus, no placement strategy was indicated.

**Our Work in Comparison.** These experiments, demonstrating the need for further analysis on prompting, were ad hoc in nature. They contained too few samples (often less than ten) to evaluate statistical significance (Cheng et al., 2023a; Hu et al., 2023; Mayladan et al., 2023), rarely explored multiple prompt types together Cheng et al. (2023a); Dai et al. (2023); Hu et al. (2023), and frequently did not give enough information to reproduce results or analyze trends, particularly with respect to how points were placed (Cheng et al., 2023a; Hu et al., 2023; Mayladan et al., 2023). In comparison, our work addresses each of these shortcomings. We offer the first large-scale evaluation of prompts, containing 2,688 distinct prompts, enabling the statistical significance of each prompt type and each prompt augmentation method to be evaluated. We performed the first systematic cross-prompting evaluation, identifying each prompt type's effect on the others. Additionally, we evaluate the effect of coarse masks, which were not explored in these previous works. Lastly, we introduce iterative refinement, testing how SAM's refinement capabilities change over multiple iterations.

## 2.3 Existing Surveys on Target Domains for SAM

Prior works, such as Ji et al. (2024), discussed SAM's performance and potential application areas. However, these works primarily reviewed *where* SAM works, and focused less on *how* to improve SAM. For example, surveys such as Zhang et al. (2023a) and Zhang et al. (2024a) extensively have cataloged SAM's usage and performance across diverse domains, including image-based tasks, human–robot interaction, and more specialized fields such as remote sensing. While these studies offer a useful overview of potential domains for SAM-based application and research, they provide little guidance on improving performance or insights into the stability of results. In contrast, our work specifically targets methods for improving performance, particularly on scientific images, a domain where SAM is known to struggle (Ma et al., 2024).

# 3 Constructing SAM Prompts

SAM was trained on the SA-1B dataset, containing 11 million images with over 1 billion masks, making it a highly capable and one of the most popular segmenters (Kirillov et al., 2023). While SAM is skilled at detecting contours, it demonstrates semantic ambiguity, occasionally assigning higher confidence to undesired sub-regions (e.g., a t-shirt instead of a person, or a sub-component of a particle as shown in Figure 1). To overcome this ambiguity, a well-designed prompt must accurately delineate the intended object, providing full coverage to prevent erroneous annotations of sub-regions, yet not so expansive that it merges the object with its neighbors or surrounding error.

When constructing a prompt, SAM takes two types of inputs: sparse (points, boxes, and text) and dense (mask), as categorized in Figure 2. Of these, boxes, points, and masks are spatial prompts, specifying exact locations or regions for SAM to process, while text prompts are non-spatial by offering semantic guidance. In our analysis, we cover all types of spatial prompts. In addition, we tested numerous prompt engineering strategies in the form of visual prompt augmentations in an effort to represent the desired object more closely. For example, we evaluated the impact of shrinking or enlarging boxes and masks. For points of interest (POIs), the augmentation steps are more complex due to their interconnected nature. Specifically, among the three augmentation factors for points (number of points, placement algorithm, and perimeter buffer), we note that the actual placement depends jointly on all three, since the buffer modifies the available search space, and the number of points augments the decision process for each algorithm.

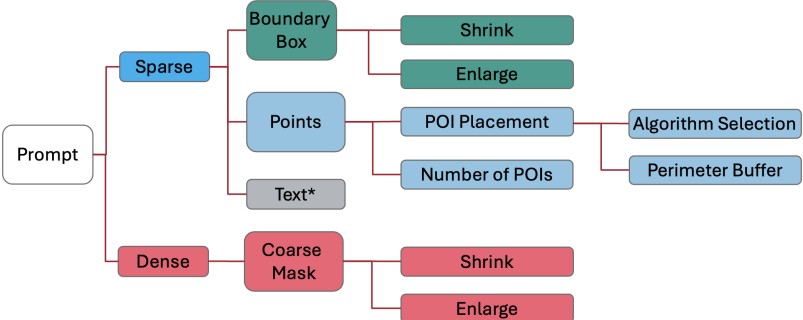

Figure 2: Characterization of prompt types and augmentations for SAM-based segmentation refinement.

## 3.1 Bounding Box Visual Prompt and Its Augmentations

Using predictions from an initial model, which we aim to refine, a bounding box $\mathbf{B}$ can be extracted for each detected instance. This box is the tightest boundary area containing the instance mask, represented by the $x$, $y$ coordinates of the upper-left corner, the width, and the height (Eq. 1). This box can be used as a prompt for SAM as-is without any augmentation. However, it can also be augmented by a scaling factor, $\alpha$, to produce $\mathbf{B}'$, as described in Equation 3. This is achieved by uniformly scaling the width and height by $\sqrt{\alpha}$, creating $w'$ and $h'$, and adjusting $x$ and $y$ by half this change, producing $x'$ and $y'$, as described in Equation 2. The resultant augmentation produces a bounding box whose area is scaled by some factor $\alpha$, while the center is held constant, preserving the spatial location of the segmentation. This augmentation can be useful in cases of poor initial segmentations, enabling expansion for under-segmentation cases or contraction for over-segmentation cases.

$$\mathbf{B} = (x, y, w, h) \tag{1}$$

$$w' = \sqrt{\alpha}\, w, \qquad h' = \sqrt{\alpha}\, h \qquad x' = x + \frac{(1 - \sqrt{\alpha})\, w}{2}, \qquad y' = y + \frac{(1 - \sqrt{\alpha})\, h}{2} \tag{2}$$

$$\mathbf{B}' = \alpha \cdot \mathbf{B} = (x', y', w', h') \tag{3}$$

## 3.2 Coarse Mask Visual Prompt and Its Augmentations

Like the bounding box, a coarse mask $\mathbf{M} \subseteq \Omega$ (where $\Omega \subset \mathbb{Z}^2$ is the set of all image-pixel coordinates) can be given to SAM as detected, or can be augmented by some factor $\lambda$ to produce $\mathbf{M}'$, an enlarged or shrunk version. However, as a complex shape, $\mathbf{M}$ cannot be uniformly scaled in the same way while preserving the spatial location and morphological characteristics of the object. Instead, $\mathbf{M}$ must be augmented at the pixel level, applying erosion ($\ominus$) to shrink and dilation ($\oplus$) to enlarge it by $\lambda$. First, we define a discrete disk with radius $t$, as shown in Equation 4. Next, define the smallest $t = r(\lambda)$ such that eroding or dilating $\mathbf{M}$ by $D_t$ yields a mask whose area is approximately $\lambda|M|$, as shown in Equation 5. Finally, we define $\mathbf{M}'$ as $\mathbf{M}$, with the inclusion of any additional pixel contained within the disk $D_t$ centered at some pixel in $\mathbf{M}$ for dilation or $\mathbf{M}$ after excluding any pixel contained within the disk $D_t$ centered at some pixel outside of $\mathbf{M}$ for erosion, as shown in Equation 6. Through this transformation, we preserve the spatial location and a similar morphology.

$$D_t \;=\; \left\{ p = (p_x, p_y) \in \mathbb{Z}^2 : \sqrt{p_x^2 + p_y^2} \;\leq\; t \right\} \tag{4}$$

$$r(\lambda) \;=\; \begin{cases} \min\left\{ t \in \mathbb{N} : \left| \mathbf{M} \oplus D_t \right| \;\geq\; \lambda\left|\mathbf{M}\right| \right\}, & \lambda \geq 1, \\[2mm] \min\left\{ t \in \mathbb{N} : \left| \mathbf{M} \ominus D_t \right| \;\leq\; \lambda\left|\mathbf{M}\right| \right\}, & 0 < \lambda < 1. \end{cases} \tag{5}$$

$$\mathbf{M}' \;=\; \begin{cases} \mathbf{M} \;\oplus\; D_{r(\lambda)} = \{ u \in \Omega : \exists\, v \in \mathbf{M},\, d \in D_{r(\lambda)} \text{ with } u = v + d \}, & \lambda \geq 1, \\[2mm] \mathbf{M} \;\ominus\; D_{r(\lambda)} \;=\; \left\{ u \in \Omega : \exists\, v \in \mathbf{M},\, d \in D_{r(\lambda)} \text{ with } u = v - d \right\}, & 0 < \lambda < 1. \end{cases} \tag{6}$$

## 3.3 POI Placement Augmentations

When evaluating POI augmentation strategies like POI placement, we aim to balance two competing interests: spacing the POIs far enough apart to fully cover the desired object, while keeping them close enough to avoid false prompts outside the object. To address this, we distinguish between and then evaluate three prompt placement parameters: the number of POIs (denoted as $k$), the perimeter exclusion area (denoted as $\gamma$), and the POI placement algorithm (denoted as $\phi$). In this work, we explore values of $k$ ranging from 1 to 7. We evaluate $\gamma$ as a mitigating factor to prevent POIs from being placed too close to the object boundary, where false prompts are more likely. In our experiments, we apply a perimeter buffer $\gamma$ of 0%, 5%, 10%, and 15%, eroding the original mask $\mathbf{M}$ to yield the search region $\mathbf{M}^*$ with Equations 4 - 6. Note the distinction between $\mathbf{M}^*$ and $\mathbf{M}'$: the parameter $\gamma$ only defines the eroded region used for POI placement, whereas $\lambda$ governs how we dilate or erode the mask input itself.

With the number of POIs $k$, and the POI search space defined by $\gamma$, we design and then evaluate three POI placement algorithms ($\phi$): random, distance maximization, and Voronoi placement. The three algorithms are described below.

**Random Placement.** For random placement (Alg. 1), the initial mask $\mathbf{M}$, perimeter buffer $\gamma$, and number of points $k$ are taken as inputs. Once $\mathbf{M}$ is eroded by $\gamma$ to produce $\mathbf{M}^*$, defining the search space, we uniformly sample $k$ points.

**Distance Maximization.** We design a second algorithm, called distance maximization, with the aim to address two issues that can arise from random placement, namely: (i) points landing too close to each other, and thus likely to add little additional value, and (ii) sub-components of an object left unprompted due to all points being concentrated elsewhere. The intuition is to maximize pairwise distances between points, we avoid point crowding, broaden spatial coverage, and reduce the likelihood that any object subregion is missed. To achieve this (Alg. 2), the algorithm accepts the same inputs. Further, we define $M^*$ the same as for random placement, and $V$ as all points contained within $M^*$. For each $C \subseteq V$ such that $|C| = k$, we evaluate the total pairwise distance between all points $p \in C$, and return $P$ as the set $C$ with the largest total pairwise distance.

---

**Algorithm 1:** Random Point Placement

---

**Input:** Binary mask $\mathbf{M}$, perimeter buffer $\gamma$, number of points $k$
**Output:** Selected points $P$
$M^* \leftarrow M \ominus (1 - \gamma)$;
$V \leftarrow \{(x, y) \in M^*\}$;
$P \leftarrow \emptyset$;
**while** $|P| < k$ **do**
    pick $(x, y)$ uniformly at random from $V$;
    $P \leftarrow P \cup \{(x, y)\}$;
    $V \leftarrow V \setminus \{(x, y)\}$;
**return** $P$;

---

**Algorithm 2:** Distance-Maximization Placement

---

**Input:** Binary mask $\mathbf{M}$, perimeter buffer $\gamma$, number of points $k$
**Output:** Selected points $P$
$M^* \leftarrow M \ominus \gamma$;
$V \leftarrow \{(x, y) \mid \text{pixel } (x, y) \text{ is on the boundary of } M^*\}$;
maxDistance $\leftarrow -\infty$;
$P \leftarrow \emptyset$;
**for** *each subset* $C \subset V$ *with* $|C| = k$ **do**
    tempDistance $\leftarrow totalPairwiseDistance(C)$;
    **if** tempDistance > maxDistance **then**
        maxDistance $\leftarrow$ tempDistance;
        $P \leftarrow C$;
**return** $P$;

---

**Voronoi Placement.** Our third algorithm, Voronoi placement, continues to avoid point crowding and to maximize coverage. However, in addition, the algorithm aims to reduce prompting outside of the true object– an issue that can arise with distance maximization. The Alg. 3 again accepts the same inputs and defines $M^*$ as above. To achieve the above goal, however, it now approximates a centroidal Voronoi tessellation of $M^*$ into $k$ cells via Lloyd's $k$-means iterations. The algorithm then returns the centroid of each cell as the POI placement.

---

**Algorithm 3:** Voronoi Point Placement

---

**Input:** Binary mask $\mathbf{M}$, perimeter buffer $\gamma$, number of points $k$, iterations $I$
**Output:** Selected points $P$
$M^* \leftarrow M \ominus \gamma$;
$V \leftarrow \{(x, y) \in M^*\}$;
Initialize $c_1, \ldots, c_k$ by sampling $k$ distinct points from $V$;
**for** $i \leftarrow 1$ **to** $I$ **do**
    **foreach** $v \in V$ **do**
        assign $v$ to cluster $j = \arg\min_i \|v - c_i\|_2$;
    **for** $i \leftarrow 1$ **to** $k$ **do**
        $c_i \leftarrow \text{mean}(\{v \in V \mid v \text{ assigned to } i\})$;
$P \leftarrow \{\text{round}(c_i)\}_{i=1}^{k}$;
**return** $P$;

---

With random placement as a baseline, Voronoi and distance maximization use the same underlying strategy for placement, assuming that points farther apart will be more representative of the complete mask. However,

since Voronoi places points at the centroid of each cluster, these points are not as likely to be placed on the boundary and will also be spread around the inner regions of the mask more, resulting in a distinct approach. Piecing this together, the set of POIs is determined by the initial mask ($\mathbf{M}$), the number of POIs ($k$), the chosen perimeter buffer ($\gamma$), and the selected algorithm for placement ($\phi$), as in Equation 7.

$$\mathbf{Ps} = f(\mathbf{M}, \phi, k, \gamma) \tag{7}$$

### 3.4 Final Prompt Composition

Ultimately, the final prompt, $\mathcal{P}$, corresponds to the combination of the desired components after augmentation, as described in Equation 8, and visualized in Figure 3. Since a prompt can contain any combination of visual prompt types, we represent the inclusion (or exclusion) of each component with $\delta_B$, $\delta_M$, and $\delta_P$ as binary flags. Here, $\delta_B$ indicates if the box $\mathbf{B}'$ is included, $\delta_M$ specifies whether the mask $\mathbf{M}'$ is included, and $\delta_P$ determines if the generated POIs, $\mathbf{Ps}$, are incorporated. As shown in Figure 3, after all necessary augmentations have been performed, the desired inputs are joined together and fed to SAM for re-segmentation, producing a final refined segmentation.

$$\mathcal{P} = \big\{ \delta_B\, \mathbf{B}',\ \delta_M\, \mathbf{M}',\ \delta_P\, \mathbf{Ps} \ \big|\ \delta_B, \delta_M, \delta_P \in \{0, 1\} \big\}. \tag{8}$$

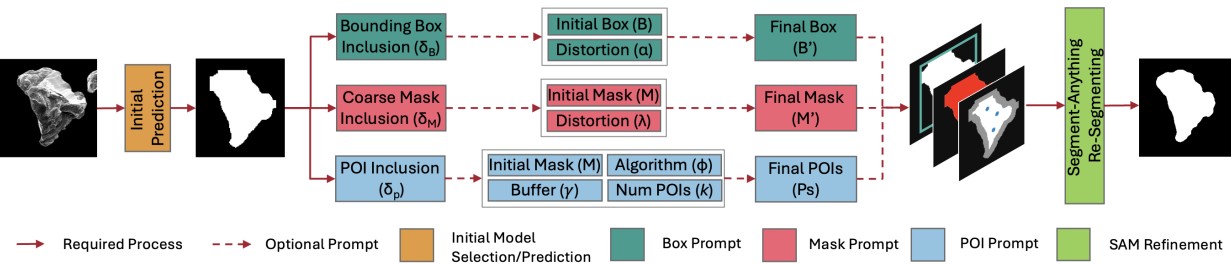

Figure 3: Framework of prompt input selection and augmentation considered for SAM-based refinement.

## 4 Experimental Methodology

### 4.1 Datasets

For the primary analysis, we used a microscopy dataset composed of micrographs from a scanning electron microscope of powder particles used in additive manufacturing. In powder-based additive manufacturing, particle size and shape are critical to overall build quality. Particles too large can leave porous regions, negatively impacting the structural integrity of manufactured parts. Similarly, particles with irregular morphology can affect the flowability of a powder, decreasing the deposition efficiency and increasing costs. As a result, there has been a growing interest in high-throughput powder analysis frameworks (Cohn et al., 2021; Price et al., 2021). For the primary evaluation, we used a powder dataset containing 132 images and 7,324 annotations (Price et al., 2025). To ensure generalizability, after identifying the optimal prompt configuration on the primary powder-particle dataset, we compared SAM's refinement performance using our identified prompt against state-of-the-art methods on two additional publicly available microscopy datasets: a general collection of 385 images covering particles, grains, and cells with 8,877 annotations (Tonneau, 2023), and a cellular microscopy dataset of 65 images with 2,193 annotations (Yuki, 2024).

### 4.2 Models

**Vision Models for Initial Segmentation.** Four vision models were used for initial segmentation: YOLOv8 Nano, YOLOv8 X-Large (Ultralytics, 2023), Mask R-CNN (He et al., 2017), and Mask2Former (Cheng et al., 2022). These models were chosen for their growing popularity in applied scientific domains

(Zhu et al., 2025; Zhang et al., 2024b;d; Tang et al., 2025; Wankhade et al., 2024; Cohn et al., 2021; Price et al., 2021). These models also cover a diverse range of model sizes and architectures, including convolutional-based and transformer-based backbones.

**Vision Models for Segmentation Refinement.** For SAM refinement, we used the ViT-L backbone (Alexey, 2020) with the published model weights and parameters, as specified in Kirillov et al. (2023). For comparison against state-of-the-art refinement techniques, we employed CascadePSP, introduced at CVPR 2020 (Cheng et al., 2020), in its default configuration. Similarly, we used SegRefiner, presented at NeurIPS 2023 (Wang et al., 2024), evaluating both its small and large variants.

### 4.3   Metrics

To evaluate performance, *Average Precision (AP)* was computed at Intersection-over-Union ($IoU$) thresholds of 0.50, 0.75, and 0.95, and the average of all thresholds between 0.50 and 0.95 (in 0.05 increments). AP@50 provides the most lenient metric, where a segmentation is considered correct if IoU $\geq$ 50%. In contrast, AP@95 provides the strictest metric, requiring the IoU $\geq$ 95% for a segmentation to be considered correct. AP@75 offers a middle ground between these metrics, while AP@50-95 uniformly summarizes performance across multiple thresholds. Measuring performance at various IoU thresholds enables us to track improvement at different stages of initial prediction accuracy, offering insight into where refinement is most useful.

### 4.4   Experimental Design

In this work, our experiments can be broken into three main categories: prompt optimization, state-of-the-art comparison, and statistical significance testing. For prompt optimization, we systematically evaluated 2,688 unique prompt configurations, encompassing combinations of points (number, placement algorithm, perimeter buffer), boxes (inclusion and scale), coarse masks (inclusion and erosion/dilation), and iterative refinement, on four initial segmentation models, as outlined in Table 1. For the state-of-the-art evaluation assessing generalizability of our proposed prompt configuration design, we compared segmentation results of our best prompt derived on our first dataset to CascadePSP (Cheng et al., 2020), SegRefiner Small and SegRefiner Large (Wang et al., 2024) on all four initial segmentation models and three microscopy datasets (powder, general microscopy, and cells). Lastly, to validate our measured improvements, we conducted paired *t*-tests on individual prompt parameters (e.g., box vs. no-box, various POI strategies, etc.) as well as overall performance differences between SAM and alternate refinement methods.

Table 1: Experimental conditions tested for prompt creation.

| Category | EQ | Conditions |
|---|---|---|
| Model | – | YOLOv8 Nano, YOLOv8 X-Large, Mask R-CNN, Mask2Former |
| Box Inclusion | $\delta_B$ | Yes / No |
| Box Distortion | $\alpha$ | 90%, 100%, 110% (when included) |
| Mask Inclusion | $\delta_M$ | Yes / No |
| Mask Distortion | $\lambda$ | 90%, 100%, 110% (when included) |
| POI Algorithm | $\phi$ | Random, Distance Max, Voronoi |
| POI Perimeter Buffer | $\gamma$ | 0%, 5%, 10%, 15% |
| Number of POIs | $k$ | 1, 2, 3, 4, 5, 6, 7 |
| Iterative Refinement | – | One Stage / Two Stage |

**Rationale for Parameter Ranges.** When designing these experiments, our objective was to improve boundary quality while preserving the location and size of an object, established by the domain-trained base model. Accordingly, we limited box and mask adjustments to $\pm 10\%$. For the perimeter buffer, we sought to reduce the likelihood of false points near uncertain edges without overpowering the placement algorithm. We thus varied perimeter buffer by $\gamma \in \{0\%, 5\%, 10\%, 15\%\}$. Lastly, we also explored the effect of increasing the number of points.However, prior work in interactive prompting with SAM has shown that the optimal

number of points was approximately seven or fewer before performance started to deteriorate with additional points (Quesada et al., 2024). Thus, in this work, we constrained $k$ to a maximum of seven points.

## 5 Experimental Results

In our prompt optimization experiments, we identified multiple trends leading to a well-designed prompt, which we discuss further in Section 5.1. From these results, we determined the optimal prompt included a bounding box (as-detected), no coarse mask, 3 POIs, Voronoi placement, and a 10% perimeter buffer, yielding crisper and more precise segmentation across all four initial segmentation models, as shown in Figure 4a. Using this prompt, on the powder dataset, we found statistically significant improvement in the AP@50-95 by 0.203, 0.201, 0.128, and 0.081 with $p < 0.0001$ for YOLOv8 Nano, YOLOv8 X-Large, Mask R-CNN, and Mask2Former, as shown in Table 2. Compared to the state-of-the-art CascadePSP, SegRefiner-Small, and SegRefiner Large, SAM consistently outperformed each on all four models.

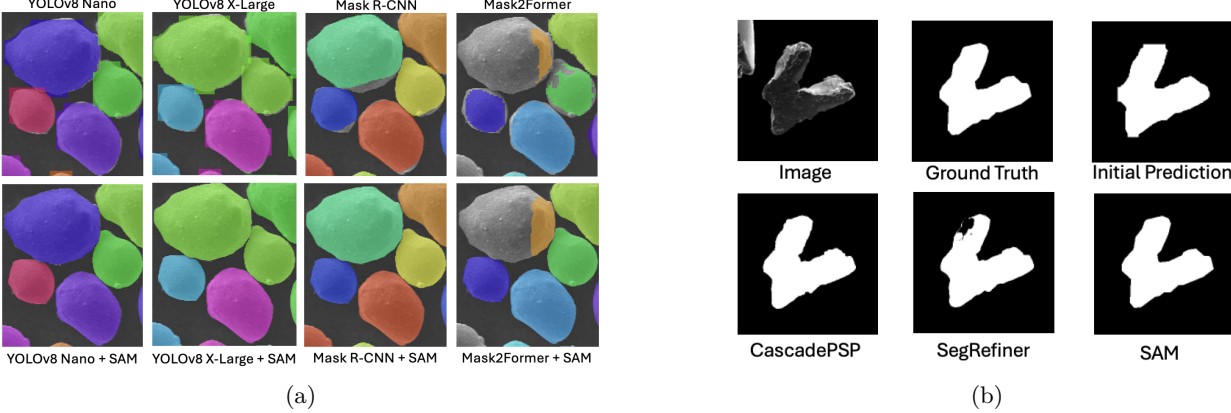

(a)   (b)

Figure 4: **(a)** Qualitative evaluation of SAM-Based refinement on YOLOv8 Nano, YOLOv8 X-Large, Mask R-CNN, and Mask2Former segmentations. **(b)** Qualitative comparison of CascadePSP, SegRefiner, and an optimized SAM prompt refining segmentation, producing an IoU of 0.960, 0.926, and 0.976, respectively.

Table 2: Paired $t$-Test of Refined vs Initial AP@50–95 by Model and Method on Powder Dataset

| Model | Method | $t$ | $p$ | Mean $\Delta$AP@50–95 | Std $\Delta$AP@50–95 | 95% CI |
|---|---|---|---|---|---|---|
| YOLOv8 Nano | CascadePSP | 33.09 | 0.0000 | 0.153 | 0.179 | 0.144–0.162 |
| | SegRefiner - Small | 5.84 | <0.0001 | 0.037 | 0.219 | 0.025–0.050 |
| | SegRefiner - Large | 7.60 | <0.0001 | 0.052 | 0.231 | 0.038–0.064 |
| | SAM (This Work) | 42.38 | <0.0001 | 0.203 | 0.186 | 0.194–0.213 |
| YOLOv8 X-Large | CascadePSP | 33.10 | <0.0001 | 0.149 | 0.175 | 0.140–0.158 |
| | SegRefiner - Small | 6.89 | <0.0001 | 0.042 | 0.208 | 0.030–0.054 |
| | SegRefiner - Large | 9.52 | <0.0001 | 0.060 | 0.214 | 0.047–0.072 |
| | SAM (This Work) | 41.19 | <0.0001 | 0.201 | 0.190 | 0.192–0.211 |
| Mask R-CNN | CascadePSP | 20.81 | <0.0001 | 0.082 | 0.153 | 0.074–0.090 |
| | SegRefiner - Small | 11.02 | <0.0001 | 0.058 | 0.205 | 0.048–0.069 |
| | SegRefiner - Large | 12.37 | <0.0001 | 0.068 | 0.215 | 0.058–0.079 |
| | SAM (This Work) | 21.82 | <0.0001 | 0.128 | 0.229 | 0.117–0.140 |
| Mask2Former | CascadePSP | 16.50 | <0.0001 | 0.059 | 0.140 | 0.052–0.067 |
| | SegRefiner - Small | 4.95 | <0.0001 | 0.020 | 0.154 | 0.012–0.027 |
| | SegRefiner - Large | 10.36 | <0.0001 | 0.036 | 0.137 | 0.030–0.044 |
| | SAM (This Work) | 19.61 | <0.0001 | 0.081 | 0.161 | 0.073–0.089 |

An ablation of these results, shown in Table 3, highlights how incrementally adding each component continues to improve performance. Starting from a single-point random-placement baseline, adding a 10% perimeter buffer, and then increasing to three points yields consistent gains across all four backbones. Switching from random to Voronoi placement provides an additional but smaller boost on three backbones, and a negligible decrease on the fourth (-0.0007). Lastly, introducing a bounding box led to the largest increase in performance, ranging from +0.0482 to +0.1248 increase on top of previous performance improvements achieved by the choice of the placement algorithm, the number of points, and the perimeter buffer.

Table 3: Prompt-construction ablation highlighting how segmentation refinement performance (mean AP@50–95) evolves as components are added. For each initial segmentation model, the columns report the mean and the change ($\Delta$) from the prior step.

| Ablation Step | YOLOv8 Nano | | YOLOv8 X-Large | | Mask R-CNN | | Mask2Former | |
|---|---|---|---|---|---|---|---|---|
| | Mean AP@50–95 | $\Delta$ | Mean AP@50–95 | $\Delta$ | Mean AP@50–95 | $\Delta$ | Mean AP@50–95 | $\Delta$ |
| Baseline: 1 Point | 0.4612 | – | 0.4999 | – | 0.4374 | – | 0.3973 | – |
| + 10% perimeter buffer | 0.4731 | +0.0118 | 0.5369 | +0.0370 | 0.4913 | +0.0539 | 0.4055 | +0.0082 |
| + Increase to 3 Points | 0.5821 | +0.1090 | 0.5530 | +0.0161 | 0.5590 | +0.0676 | 0.4673 | +0.0619 |
| + Voronoi placement | 0.5909 | +0.0088 | 0.5987 | +0.0457 | 0.5582 | -0.0007 | 0.4686 | +0.0012 |
| + Add Bounding Box | 0.7157 | +0.1248 | 0.6880 | +0.0893 | 0.6497 | +0.0915 | 0.5167 | +0.0482 |

Table 4: Performance comparison of refinement methods across multiple initial model predictions for the Powder, General Microscopy, and Biological Cells from Scanning Electron Microscope Images. The best result in each group is highlighted in **bold**.

| Model | Powder | | | | General Microscopy | | | | Cells | | | |
|---|---|---|---|---|---|---|---|---|---|---|---|---|
| | $AP^{50}_{mask}$ | $AP^{75}_{mask}$ | $AP^{95}_{mask}$ | $AP^{50:95}_{mask}$ | $AP^{50}_{mask}$ | $AP^{75}_{mask}$ | $AP^{95}_{mask}$ | $AP^{50:95}_{mask}$ | $AP^{50}_{mask}$ | $AP^{75}_{mask}$ | $AP^{95}_{mask}$ | $AP^{50:95}_{mask}$ |
| *YOLOv8 Nano* | *0.8373* | *0.6693* | *0.0806* | *0.5291* | *0.8959* | *0.7866* | *0.2753* | *0.6526* | *0.5630* | *0.1927* | *0.0000* | *0.2519* |
| + CascadePSP | 0.8450 | 0.7529 | 0.3781 | 0.6587 | 0.8910 | 0.7878 | **0.3427** | **0.6738** | 0.5061 | 0.4105 | 0.0002 | 0.3056 |
| + SegRefiner - Small | 0.8078 | 0.6965 | 0.1843 | 0.5629 | 0.7174 | 0.6381 | 0.2099 | 0.5218 | 0.3375 | 0.1291 | 0.0000 | 0.1555 |
| + SegRefiner - Large | 0.8344 | 0.7267 | 0.3455 | 0.6355 | 0.7476 | 0.6784 | 0.3020 | 0.5760 | 0.5245 | 0.2924 | 0.0000 | 0.2723 |
| + SAM (This Work) | **0.8812** | **0.8055** | **0.4591** | **0.7153** | **0.8930** | **0.8193** | 0.2300 | 0.6474 | **0.5922** | **0.4249** | **0.0016** | **0.3395** |
| *YOLOv8 X-Large* | *0.8352* | *0.6841* | *0.0964* | *0.5386* | *0.8817* | *0.7633* | *0.2134* | *0.6195* | *0.5720* | *0.2152* | *0.0000* | *0.2624* |
| + CascadePSP | 0.8446 | 0.7587 | 0.3870 | 0.6635 | 0.8851 | 0.7880 | **0.2764** | **0.6498** | 0.5061 | 0.3986 | **0.0025** | 0.3024 |
| + SegRefiner - Small | 0.8105 | 0.6865 | 0.1860 | 0.5610 | 0.7033 | 0.6205 | 0.2279 | 0.5172 | 0.4035 | 0.1250 | 0.0000 | 0.1762 |
| + SegRefiner - Large | 0.8329 | 0.7375 | 0.3523 | 0.6409 | 0.7188 | 0.6375 | 0.2702 | 0.5421 | 0.5307 | 0.1873 | 0.0016 | 0.2399 |
| + SAM (This Work) | **0.8775** | **0.8102** | **0.4757** | **0.7211** | **0.8959** | **0.8132** | 0.2120 | 0.6404 | **0.5556** | **0.4038** | 0.0000 | **0.3198** |
| *Mask R-CNN* | *0.6876* | *0.5599* | *0.1061* | *0.4512* | *0.6653* | *0.5876* | *0.0502* | *0.4344* | *0.3438* | *0.1394* | *0.0000* | *0.1611* |
| + CascadePSP | 0.7254 | 0.6305 | 0.3245 | 0.5601 | 0.6870 | 0.5833 | 0.1775 | 0.4826 | 0.3519 | 0.1596 | 0.0000 | 0.1705 |
| + SegRefiner - Small | 0.6899 | 0.6050 | 0.2056 | 0.5001 | 0.6952 | 0.6196 | 0.2062 | 0.5070 | 0.2485 | 0.0761 | 0.0000 | 0.1082 |
| + SegRefiner - Large | 0.7291 | 0.6434 | 0.3031 | 0.5586 | 0.7176 | 0.6317 | **0.2698** | 0.5397 | **0.3548** | **0.1754** | **0.0079** | **0.1794** |
| + SAM (This Work) | **0.7788** | **0.7130** | **0.4692** | **0.6537** | **0.8959** | **0.8132** | 0.2120 | **0.6404** | 0.3289 | 0.1584 | 0.0000 | 0.1624 |
| *Mask2Former* | *0.6336* | *0.5111* | *0.1229* | *0.4225* | *0.8539* | *0.7537* | *0.2852* | *0.6309* | *0.3795* | *0.1873* | *0.0001* | *0.1890* |
| + CascadePSP | **0.6673** | **0.5805** | 0.2387 | 0.4955 | **0.8693** | 0.7337 | **0.2909** | **0.6313** | 0.3584 | 0.2258 | **0.0007** | 0.1950 |
| + SegRefiner - Small | 0.6300 | 0.4871 | 0.1176 | 0.4116 | 0.6777 | 0.5752 | 0.1847 | 0.4792 | 0.2626 | 0.0631 | 0.0000 | 0.1086 |
| + SegRefiner - Large | 0.6626 | 0.5718 | 0.2357 | 0.4900 | 0.7340 | 0.6218 | 0.2675 | 0.5411 | **0.3706** | 0.1146 | <0.0001 | 0.1617 |
| + SAM (This Work) | 0.6548 | 0.5497 | **0.3132** | **0.5059** | 0.8588 | **0.7827** | 0.2003 | 0.6139 | 0.3533 | **0.2620** | 0.0006 | **0.2053** |

Extending this to additional AP thresholds, as shown in Table 4, we found that SAM achieved the highest AP@95 across all four base segmentation models. At AP@50 and AP@75, SAM outperformed both SegRefiner Small and SegRefiner Large on every initial segmentation model, and exceeded CascadePSP on three of the four models. Expanding this analysis to the two additional datasets, general microscopy and cellular microscopy, as shown in Table 4, we found that SAM continued to improve segmentation quality, and perform comparably or better to the state-of-the-art methods. For AP@75, scored the highest on all four initial segmentation models for both datasets. For AP@50, SAM scored the best in four of eight measurements, and second best in an additional two.

> **Key Takeaway 1:** Using a prompt with a bounding box, 3 points, Voronoi placement, a 10% perimeter buffer, and no coarse mask, SAM produced statistically significant improvement over the raw segmentations, and performed consistently better than or comparably to CascadePSP or SegRefiner across three microscopy datasets - illustrating the generalizability of prompts for these scientific image datasets.

## 5.1 Prompt Optimization

### 5.1.1 Impact of Bounding Boxes on Prompt Performance

Of the 2,688 prompts tested, these were evenly split between no bounding box, as-detected bounding box, bounding box shrunk to 90% of its original size, and bounding box enlarged to 110% of its original size. To evaluate impact, paired $t$-tests were conducted, comparing each box configuration to an identical prompt without a box, as shown in Table 5. On average, including an as-detected bounding box had an average AP@50-95 improvement of 0.32. Shrinking this box to 90% of its size had a slightly larger improvement at 0.341, while enlarging the box had a slightly smaller improvement at 0.293.

Table 5: Paired $t$-Tests on Impact of Bounding Box Scale on Segmentation Refinement (AP@50–95)

| Box Scale | $t$ | $p$ | Mean $\Delta$AP | Std $\Delta$AP | 95% CI |
|---|---|---|---|---|---|
| 90% | 105.77 | <0.0001 | 0.341 | 0.167 | 0.334–0.347 |
| 100% | 106.39 | <0.0001 | 0.320 | 0.156 | 0.314–0.326 |
| 110% | 103.99 | <0.0001 | 0.293 | 0.146 | 0.287–0.298 |

### 5.1.2 Impact of Coarse Masks on Prompt Performance

Contrary to bounding boxes, including coarse masks in prompts consistently negatively affected prompt performance in our study. Broken down by mask augmentation, 90%, 100% (as-detected), and 110%, paired $t$-tests revealed an average decrease of -0.133 AP@50-95, as shown in Table 6. Interestingly, mask augmentation was found to have no impact, with each having an average decrease of -0.133, an STD of 0.137, and a 95% confidence interval of -0.138– -0.128. In the future, it would be beneficial to evaluate whether this trend persists on natural images, which may feature textured backgrounds, occlusions, and thin or elongated structures that could alter the utility of coarse masks.

Table 6: Paired $t$-Tests on Impact of Coarse Mask Scale on Segmentation Refinement (AP@50–95)

| Mask Scale | $t$ | $p$ | Mean $\Delta$AP | Std $\Delta$AP | 95% CI |
|---|---|---|---|---|---|
| 90% | -50.38 | <0.0001 | -0.133 | 0.137 | -0.138– -0.128 |
| 100% | -50.47 | <0.0001 | -0.133 | 0.137 | -0.138– -0.128 |
| 110% | -50.38 | <0.0001 | -0.133 | 0.137 | -0.138– -0.128 |

### 5.1.3 Combined Effect of Bounding Boxes and Coarse Masks

In addition to the independent effects of bounding boxes and coarse masks on performance, we found that their combined impact is even more pronounced. For example, prompts that include a bounding box outperform those that omit it, while prompts that include a coarse mask underperform those that exclude it. Further, prompts featuring a box without a mask yield the highest average performance, whereas prompts with a mask but no box yield the lowest, as shown in Figure 5A.

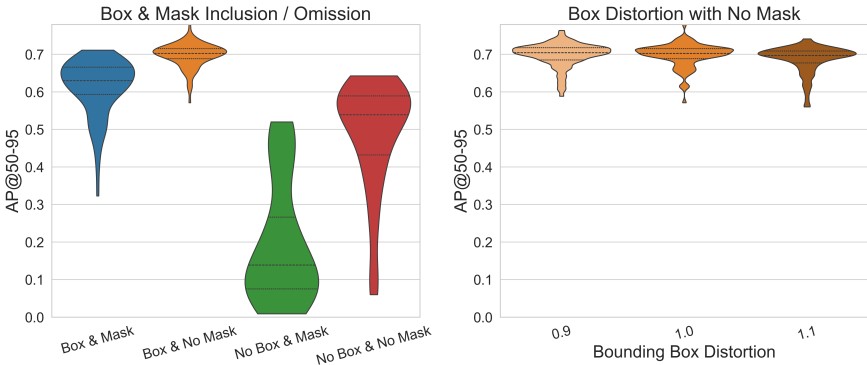

Figure 5: Visualizing impact of bounding boxes and masks on AP@50-95 when using SAM-based refinement.

---

**Key Takeaway 2:** Prompts that included a bounding box improved AP@50–95 by an average of 0.32 compared to identical prompts without a box, whereas prompts that included a coarse mask decreased AP@50–95 by an average of 0.133. Overall, the highest performance was achieved by prompts that included a bounding box and excluded a coarse mask.

---

Focusing on the best of these combinations (yes box and no mask), Figure 5B highlights the distribution of AP@50-95 scores when the bounding box size was shrunk to 90%, or enlarged to 110% of its original size. Minor differences in distributions can be observed, such as a slightly lower tail distribution for prompts with an enlarged box (0.560 compared to 0.571) and a slightly smaller distribution when the shrunk bounding boxes (0.175 compared to 0.209). However, we conclude that overall box augmentation had no significant effects on performance.

## 5.2 POI Placement Impact

Analyzing the effect of POI prompt placement, we considered the individual and combined effect of the number of points ($k$), POI placement algorithm ($\phi$), and perimeter buffer ($\gamma$). In addition, we evaluated the effect of these parameters in the presence and absence of a bounding box, allowing us to quantify how box inclusion modulates the influence of POI placement strategies on segmentation refinement.

### 5.2.1 Algorithm Selection ($\phi$)

To analyze the impact of alternate POI placement algorithms, we treated random placement as the baseline, and conducted paired $t$-tests, measuring the change in AP@50-95 as a result of switching to distance maximization or Voronoi placement. For this analysis, we performed two sets of paired $t$-tests: one over the subset of prompts without a bounding box or mask (Table 7), and another over the subset of all prompts with a bounding box but no mask (Table 8). As shown in Table 7, Voronoi performed the best when no box was included, with a mean AP@50-95 improvement of 0.007 over random. In contrast, distance maximization performed the worst with an average decrease in performance of -0.146. In our tests on prompts including a bounding box, the mean AP@50–95 increased significantly for all three algorithms. Random increased from 0.541 to 0.645, Distance Max increased from 0.392 to 0.622, and Voronoi increased from 0.548 to 0.644. Additionally, when a box was included, as shown in Table 8, the effect of the placement algorithm diminished. For example, the performance decrease due to using Distance Max narrowed from –0.146 to –0.023. Similarly, the previously statistically significant improvement of 0.007 AP@50–95 for Voronoi over Random ($p < 0.0001$) became a non-significant decrease of –0.001 ($p = 0.1657$).

Table 7: Paired *t*-Tests Comparing Placement Algorithms (No Box, No Mask)

| Comparison | $t$ | $p$ | Mean $\Delta$AP | Std $\Delta$AP | 95% CI |
|---|---|---|---|---|---|
| Random→Distance Max | -20.17 | <0.0001 | -0.146 | 0.108 | -0.160—0.132 |
| Random→Voronoi | 5.30 | <0.0001 | 0.007 | 0.020 | 0.005–0.010 |

Table 8: Paired *t*-Tests Comparing Placement Algorithms (Box, No Mask)

| Comparison | $t$ | $p$ | Mean $\Delta$AP | Std $\Delta$AP | 95% CI |
|---|---|---|---|---|---|
| Random→Distance Max | -18.93 | <0.0001 | -0.023 | 0.031 | -0.025—0.021 |
| Random→Voronoi | -1.39 | 0.1657 | -0.001 | 0.012 | -0.002–0.000 |

> **Key Takeaway 3:** In this study, the placement of POIs along the boundary of an initial detection performed significantly worse than random or Voronoi placement, particularly when no bounding box was provided. However, when a bounding box was provided, the gap between algorithm selections was substantially reduced.

### 5.2.2 Number of Points ($k$)

Depending on the algorithm and the number of points ($k$), we note that points could be placed in drastically different ways, as depicted in Figure 6. To test this effect, specifically related to the number of points, we separated prompts by placement algorithm and box inclusion. Without a bounding box, adding more POIs yielded clear gains for random (up to +0.091 AP@50–95, $p < 0.0001$) and Voronoi (up to +0.083 AP@50–95, $p < 0.0001$), while Distance Max steadily lost performance (down to -0.105, $p < 0.0001$), as shown in Table 9). However, when a bounding box was included, these trends were reversed. Random and Voronoi went from a steady increase to a slight decrease in performance, and Distance Max's decrease in performance was reduced from -0.105 to -0.044.

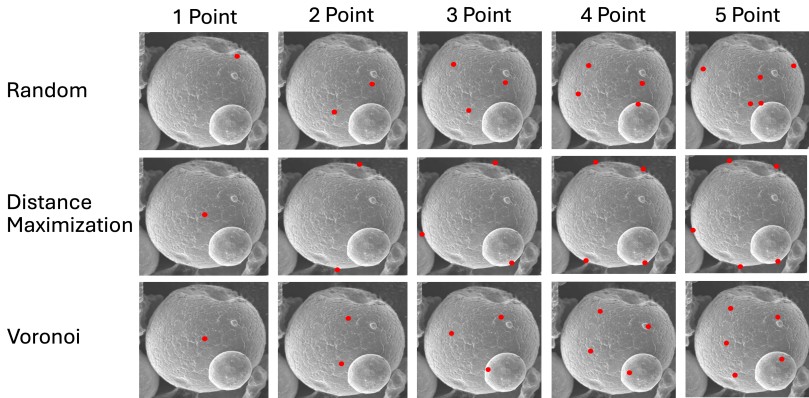

Figure 6: Sample placement of points using Random, Distance Maximization, and Voronoi placement from $k = 1$ to $k = 5$ points.

> **Key Takeaway 4:** In this study, including additional POIs improved performance for random and Voronoi placement, but decreased performance for Distance Max. However, these effects were reduced once a box was included.

Table 9: Paired $t$-Tests of Increasing POI Count (1 vs k) on AP@50–95 (No Box, No Mask)

| Algorithm | Comparison | $t$ | $p$ | Mean $\Delta$AP | Std $\Delta$AP | 95% CI |
|---|---|---|---|---|---|---|
| | 1→2 | 7.13 | <0.0001 | 0.035 | 0.027 | 0.025–0.044 |
| | 1→3 | 14.76 | <0.0001 | 0.063 | 0.024 | 0.055–0.071 |
| | 1→4 | 18.49 | <0.0001 | 0.079 | 0.024 | 0.070–0.087 |
| Random | 1→5 | 19.93 | <0.0001 | 0.089 | 0.025 | 0.081–0.098 |
| | 1→6 | 21.66 | <0.0001 | 0.091 | 0.024 | 0.083–0.099 |
| | 1→7 | 16.40 | <0.0001 | 0.090 | 0.031 | 0.079–0.100 |
| | 1→2 | -4.49 | 0.0001 | -0.052 | 0.065 | -0.077—0.032 |
| | 1→3 | -6.06 | <0.0001 | -0.082 | 0.076 | -0.111—0.059 |
| | 1→4 | -6.77 | <0.0001 | -0.093 | 0.078 | -0.121—0.069 |
| Distance Max | 1→5 | -6.70 | <0.0001 | -0.098 | 0.083 | -0.128—0.073 |
| | 1→6 | -6.86 | <0.0001 | -0.103 | 0.085 | -0.134—0.077 |
| | 1→7 | -7.02 | <0.0001 | -0.105 | 0.085 | -0.137—0.079 |
| | 1→2 | 10.40 | <0.0001 | 0.022 | 0.012 | 0.018–0.026 |
| | 1→3 | 16.12 | <0.0001 | 0.053 | 0.019 | 0.047–0.059 |
| | 1→4 | 20.95 | <0.0001 | 0.072 | 0.019 | 0.065–0.078 |
| Voronoi | 1→5 | 17.65 | <0.0001 | 0.077 | 0.025 | 0.068–0.085 |
| | 1→6 | 19.71 | <0.0001 | 0.082 | 0.024 | 0.074–0.090 |
| | 1→7 | 25.98 | <0.0001 | 0.083 | 0.018 | 0.076–0.089 |

Table 10: Paired $t$-Tests of Increasing POI Count (1 vs k) on AP@50–95 (Box Only, No Mask)

| Algorithm | Comparison | $t$ | $p$ | Mean $\Delta$AP | Std $\Delta$AP | 95% CI |
|---|---|---|---|---|---|---|
| | 1→2 | -2.11 | 0.0372 | -0.003 | 0.012 | -0.005—0.000 |
| | 1→3 | -3.53 | 0.0006 | -0.004 | 0.012 | -0.007—0.002 |
| | 1→4 | -8.03 | <0.0001 | -0.009 | 0.011 | -0.011—0.007 |
| Random | 1→5 | -8.85 | <0.0001 | -0.011 | 0.012 | -0.013—0.008 |
| | 1→6 | -10.40 | <0.0001 | -0.012 | 0.012 | -0.015—0.010 |
| | 1→7 | -10.58 | <0.0001 | -0.013 | 0.012 | -0.016—0.011 |
| | 1→2 | -7.84 | <0.0001 | -0.014 | 0.018 | -0.018—0.011 |
| | 1→3 | -10.60 | <0.0001 | -0.021 | 0.020 | -0.025—0.018 |
| | 1→4 | -9.26 | <0.0001 | -0.034 | 0.036 | -0.041—0.027 |
| Distance Max | 1→5 | -10.30 | <0.0001 | -0.042 | 0.039 | -0.050—0.034 |
| | 1→6 | -12.38 | <0.0001 | -0.045 | 0.036 | -0.052—0.038 |
| | 1→7 | -13.11 | <0.0001 | -0.044 | 0.033 | -0.051—0.038 |
| | 1→2 | -0.73 | 0.4689 | -0.001 | 0.011 | -0.003–0.001 |
| | 1→3 | -5.40 | <0.0001 | -0.005 | 0.009 | -0.007—0.003 |
| | 1→4 | -4.56 | <0.0001 | -0.006 | 0.014 | -0.009—0.004 |
| Voronoi | 1→5 | -8.91 | <0.0001 | -0.009 | 0.010 | -0.011—0.007 |
| | 1→6 | -9.77 | <0.0001 | -0.013 | 0.013 | -0.016—0.010 |
| | 1→7 | -10.07 | <0.0001 | -0.013 | 0.013 | -0.015—0.010 |

### 5.2.3 Perimeter Buffer ($\gamma$)

Based on the theory that POIs closer to the edge have a higher chance of being outside the ground truth, and therefore degrade performance, we implemented a perimeter buffer where POIs could not be placed. To evaluate this, we tested a 0% buffer (no buffer) and a 5%, 10%, and 15% buffer. When no box was included, random placement had no significant change at a 5% buffer (+0.002 AP@50-95, $p = 0.3854$),

and a small improvement at 10% (+0.008, $p = 0.0017$) and 15% (+0.012, $p = 0.0002$). Similarly, Voronoi placement had no significant change at a 5% buffer (+0.004, $p = 0.0654$), and a small improvement at 10% (+0.007, $p = 0.007$) and 15% (+0.006, $p = 0.0064$). However, Distance Max had a much larger and more statistically significant improvement at 5% (+0.053, $p < 0.0001$), 10% (+0.093, $p < 0.0001$), and 15% (+0.093, $p < 0.0001$). When evaluating prompts with a bounding box, we observed a similar trend reversal that was found in the placement algorithm selection and the number of POI selections. Random and Voronoi had no statistically significant change in AP@50-95. The improvements for Distance Max dropped from +0.053 to +0.007 at 5%, from +0.093 to +0.006 at 10%, and from +0.093 to -0.024 at 15%.

Table 11: Effect of Perimeter Buffer on AP@50–95 (No Box, No Mask)

| Algorithm | Buffer | $t$ | $p$ | Mean $\Delta$AP | Std $\Delta$AP | 95% CI |
|---|---|---|---|---|---|---|
| Random | 0%→5% | 0.87 | 0.3854 | 0.002 | 0.021 | -0.003–0.008 |
| | 0%→10% | 3.30 | 0.0017 | 0.008 | 0.019 | 0.003–0.013 |
| | 0%→15% | 4.02 | 0.0002 | 0.012 | 0.022 | 0.006–0.017 |
| Distance Max | 0%→5% | 5.27 | <0.0001 | 0.053 | 0.076 | 0.035–0.073 |
| | 0%→10% | 6.54 | <0.0001 | 0.093 | 0.107 | 0.066–0.122 |
| | 0%→15% | 5.32 | <0.0001 | 0.093 | 0.131 | 0.060–0.128 |
| Voronoi | 0%→5% | 1.89 | 0.0645 | 0.004 | 0.017 | -0.000–0.009 |
| | 0%→10% | 2.84 | 0.0063 | 0.007 | 0.018 | 0.002–0.011 |
| | 0%→15% | 2.84 | 0.0064 | 0.006 | 0.017 | 0.002–0.011 |

Table 12: Effect of Perimeter Buffer on AP@50–95 (Box Only, No Mask)

| Algorithm | Buffer | $t$ | $p$ | Mean $\Delta$AP | Std $\Delta$AP | 95% CI |
|---|---|---|---|---|---|---|
| Random | 0%→5% | 1.60 | 0.1118 | 0.001 | 0.011 | -0.000–0.003 |
| | 0%→10% | 1.46 | 0.1475 | 0.002 | 0.014 | -0.001–0.004 |
| | 0%→15% | 2.55 | 0.0118 | 0.002 | 0.012 | 0.001–0.004 |
| Distance Max | 0%→5% | 4.71 | <0.0001 | 0.007 | 0.018 | 0.004–0.009 |
| | 0%→10% | 4.16 | 0.0001 | 0.006 | 0.020 | 0.003–0.009 |
| | 0%→15% | -8.04 | <0.0001 | -0.025 | 0.040 | -0.031—0.019 |
| Voronoi | 0%→5% | -1.33 | 0.1869 | -0.001 | 0.011 | -0.003–0.000 |
| | 0%→10% | 0.86 | 0.3912 | 0.001 | 0.010 | -0.001–0.002 |
| | 0%→15% | 0.42 | 0.6772 | 0.000 | 0.011 | -0.001–0.002 |

> **Key Takeaway 5:** When no bounding box is provided, introducing a small perimeter buffer (10–15%) around the mask edge improved AP@50-95 scores, particularly for Distance Max.

**Connecting the Dots.** To visualize the interconnected nature of these parameters, Figure 7 highlights the effect of placement algorithm ($\phi$), number of points ($k$), perimeter buffer ($\gamma$), box inclusion across the four initial segmentation models. Each row shows one placement strategy (Distance Max on top, Random in the middle, Voronoi on bottom), and each column corresponds to an initial segmentation model (YOLOv8 Nano, YOLOv8 X-Large, Mask R-CNN, Mask2Former, left to right). Within each graph, an increase along the X-axis indicates a growing perimeter buffer ($\gamma$), while an increase along the Y-axis indicates an increase in reported AP@50-95. Each colored line represents the number of POIs used ($k$), with solid lines indicating a bounding box was included and dashed lines indicating the bounding box was excluded.

In these results, we confirm previous findings that bounding boxes improved results with every single prompt, performing better with a bounding box than its counterpart without a bounding box.

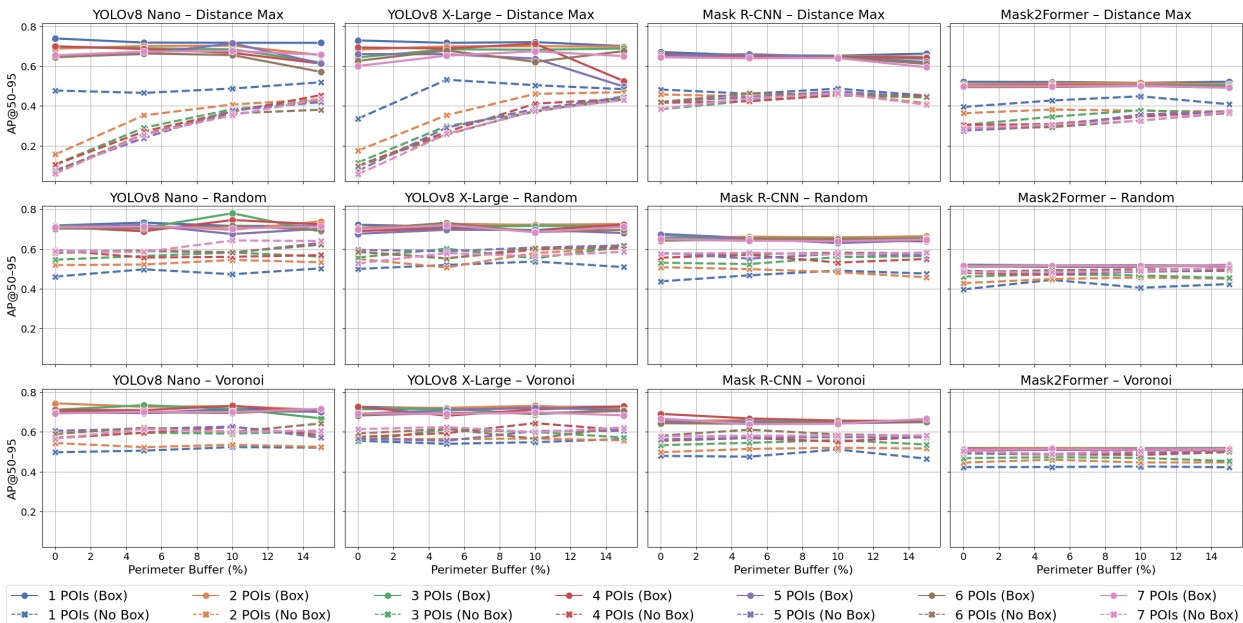

Figure 7: Evaluation of bounding box inclusion, POI placement algorithms, number of POIs, and perimeter buffer across multiple instance segmentation architectures.

## 5.3 Iterative Refinement

To evaluate whether any prompt could further improve an already refined mask, we implemented a two-stage protocol. In **Stage 1**, for each base segmentation model (see Section 4.2 and Table 1), we applied each prompt configurations $\{\mathcal{P}_i\}$ to the raw model outputs and selected the top-performing prompt $\mathcal{P}^*$. In **Stage 2**, we first used $\mathcal{P}^*$ to generate once-refined masks under optimal prompting conditions, then re-ran SAM with each prompt $\mathcal{P}_i$ on those once-refined masks to produce twice-refined outputs. We measured AP@50–95 for every $\mathcal{P}_i$ in Stage 2 and compared it to the corresponding Stage 1 score to see if any prompt yielded additional gains.

As shown in Figure 8A, the range of twice-refined AP@50-95 scores was much narrower than once-refined (Fig. 5), particularly for prompts including a bounding box, or no bounding box and no coarse mask. For example, the range of once-refined AP@50-95 scores for prompts containing a box and mask decreased from 0.39 to 0.18, prompts containing a box and no mask decreased from 0.21 to 0.10, and prompts containing neither a box nor mask dropped from 0.58 to 0.22. While we saw a significant reduction in the spread of recorded values, this was from a rising tail value, and the maximum recorded value actually decreased from once-refined to twice-refined segmentations for each group. In fact, for all $\{\mathcal{P}_i\}$, 59.24% achieved higher AP@50–95 when run on the once-refined $\mathcal{P}^*$ masks compared to the raw outputs. However, none of the twice-refined outputs from any $\mathcal{P}_i$ produced a closer segmentation to the ground truth than the once-refined mask produced by $\mathcal{P}^*$.

To analyze this further, paired $t$-tests comparing once-refined and twice-refined AP@50–95 were conducted, as shown in Table 13. For improved clarity, prompts were divided into five groups according to their once-refined AP@50–95 percentiles: 0–20%, 20–40%, 40–60%, 60–80%, and 80–100%. Prompts in the 0-20% group experienced the largest improvement (+0.034 AP@50-95, $p < 0.0001$). As the percentile group increased, the improvement decreased. For example, 20-40% had a smaller improvement (+0.008 AP@50-95, $p < 0.0001$), and 40-60% had an even smaller improvement (+0.003 AP@50-95, $p < 0.0001$). Prompts in the 60-80% range had an average improvement of 0.000 with no statistical significance ($p = 0.4481$), and prompts within the upper 80th percentile had a statistically significant decrease in performance (-0.006 AP@50-95, $p < 0.0001$).

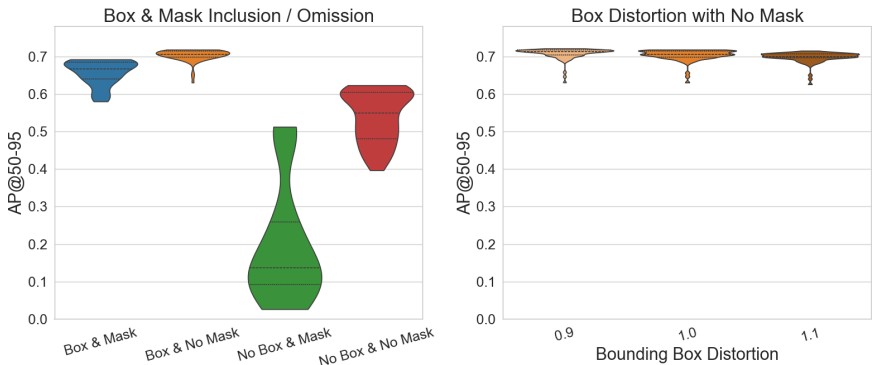

Figure 8: Impact of Box and Mask Prompts on Secondary Refinement

Table 13: Paired $t$-Test of Once-Refined vs Twice Refined Segmentations

| Percentile | $t$ | $p$ | Mean $\Delta$AP | Std $\Delta$AP | 95% CI |
|---|---|---|---|---|---|
| 0%–20% | 19.05 | <0.0001 | 0.034 | 0.059 | $0.031 - 0.038$ |
| 20%–40% | 12.23 | <0.0001 | 0.008 | 0.022 | $0.007 - 0.010$ |
| 40%–60% | 5.89 | <0.0001 | 0.003 | 0.016 | $0.002 - 0.004$ |
| 60%–80% | 0.76 | 0.4481 | 0.000 | 0.015 | $-0.001 - 0.001$ |
| 80%–100% | -14.00 | <0.0001 | -0.006 | 0.015 | $-0.007 - -0.006$ |

**Key Takeaway 6:** Prompts that initially performed quite poorly did improve when using a once-refined segmentation. However, no twice-refined segmentation outperformed the highest-scoring once-refined segmentation, and high-performing prompts consistently performed worse twice-refined than once-refined.

## 5.4 Evaluating Generalizability

While this work provides an extensive evaluation of prompt construction of SAM on scientific images, we also tested its generalizability under two alternate conditions. First, we tested if our empirically determined optimal prompt construction still yielded similar performance on SAM2, with SAM2 a newer SAM architecture designed for videos that is meant to continue to also provide image-segmentation capabilities. Further, we also tested this same prompt using SAM in a domain distinct from scientific images to evaluate its generalizability.

### 5.4.1 Alternative SAM Architectures: SAM vs. SAM2

SAM2 released in 2024 Ravi et al. (2024), one year after SAM (Kirillov et al., 2023), was designed to not only handle the segmentation of still images but now also of videos. In this study, we applied the same prompt (3 points with Voronoi placement, a 10% perimeter buffer, and a bounding box) on all four SAM2 variants (base+, tiny, small, and large), as shown in Table 14. Compared to the unrefined mask, each architecture improved performance with AP@50 increasing from 0.8469 to 0.8717-0.8748, AP@75 increasing from 0.6768 to 0.7521-0.7660, AP@95 increasing from 0.0912 to 0.3212-0.3672, and AP@50-95 increasing from 0.5383 to 0.6488-0.6670. While improving over the unrefined baseline, all SAM2 variants showed slightly lower performance than SAM on single-image refinement, with the gap most pronounced at stricter thresholds. For example, SAM2-Large reached AP@50 of 0.8722 versus 0.8806 for SAM, but at AP@95 obtained 0.3339 versus 0.4712. This slight decrease in performance is consistent with previous findings in the literature that SAM2 is not as good at fine-grained detail for images (Pei et al., 2024).

Table 14: Comparing SAM to SAM2 segmentation refinement performance on the powder dataset using 3 points, Voronoi placement, a 10% perimeter buffer, and a bounding box.

| Method | AP@50 | AP@75 | AP@95 | AP@50-95 |
|---|---|---|---|---|
| Unrefined (initial) | 0.8469 | 0.6768 | 0.0912 | 0.5383 |
| SAM2-Base+ | 0.8717 | 0.7621 | 0.3672 | 0.6670 |
| SAM2-Tiny | 0.8748 | 0.7660 | 0.3287 | 0.6565 |
| SAM2-Small | 0.8731 | 0.7521 | 0.3213 | 0.6488 |
| SAM2-Large | 0.8722 | 0.7656 | 0.3339 | 0.6572 |
| SAM (This Work) | 0.8806 | 0.7954 | 0.4712 | 0.7157 |

### 5.4.2 Alternative Image Domains

To test generalizability, we evaluated our recommended prompt configuration on natural-scene datasets in contrast of our target scientific image datasets. For this, we selected *ECSSD* (Shi et al., 2015), a collection of 1,000 natural images with textured backgrounds, specifically designed to increase segmentation difficulty. Using three initial segmentation models (RMBG-2 (BRIA AI, 2025), BiRefNet-DIS (Zheng et al., 2024), and RMBG-1.4 (BRIA AI, 2024)), we compared the intersection-over-union (IoU) of refined segmentations to their initial segmentation. Averaged across the three models, SAM (with a prompt of 3 points, Voronoi placement, a 10% perimeter buffer, and a bounding box) improved IoU by +1.79, as shown in Table 15. In comparison, CascadePSP produced an average IoU gain of +0.62, while SegRefiner-LR and SegRefiner-HR both degraded performance. These results on a natural-scene dataset indicate that the proposed prompt configuration can transfer beyond microscopy and functions as a general, model-agnostic refiner.

Table 15: Average IoU across three initial segmentation models, RMBG-2 (BRIA AI, 2025), BiRefNet (Zheng et al., 2024), and RMBG-1.4 (BRIA AI, 2024) on the ECSSD dataset (Shi et al., 2015).

| Method | RMBG-2 | BiRefNet-DIS | RMBG-1.4 | Mean |
|---|---|---|---|---|
| Unrefined (initial) | 86.85 | 79.30 | 78.09 | 81.41 |
| CascadePSP | 86.39 | 81.09 | 78.61 | 82.03 |
| SegRefiner–LR | 70.25 | 67.27 | 65.31 | 67.61 |
| SegRefiner–HR | 68.50 | 66.40 | 64.39 | 66.43 |
| SAM (This Work) | 87.78 | 83.05 | 78.77 | 83.20 |

### 5.5 Limitations

While this study demonstrates statistically significant results showing that SAM can perform similarly to or better than state-of-the-art segmentation refinement techniques, it does have a few limitations. A primary limitation is the dependence on the quality and completeness of the initial masks. Because SAM's refinement leverages the spatial support and appearance cues present in the initial segmentation, large misalignments or missing instances limit its ability to effectively refine object boundaries. This is highlighted in Figure 9, where the same prompts are used to refine the same image, except that we use initial segmentations from YOLOv8 Nano (with 76 detected objects) and Mask2Former (with 64 detected objects). SAM successfully refines many of the detected objects, improving AP@50–95 from 0.8048 to 0.9307 for YOLOv8 Nano and from 0.6560 to 0.7787 for Mask2Former. However, despite using the same prompt construction, the same SAM architecture, and the same image, the refined Mask2Former result is significantly lower. This occurs because SAM is strictly a refiner in this setting and has no way to recover missed segmentations (marked with a red X).

Second, SAM's performance is sensitive to "false POIs" (points placed outside the true object), which can rapidly deteriorate mask refinement. We mitigate this through a perimeter buffer, preventing POIs from being placed near the segmentation boundary, where they are most likely to be incorrect. However, it remains

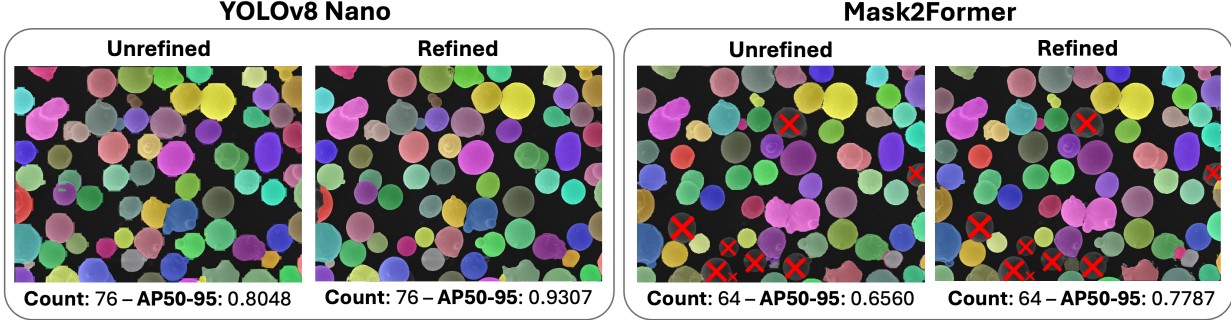

Figure 9: Example of SAM's dependence on initial segmentations. YOLOv8 Nano (left) and Mask2Former (right) on the same image show increased performance after refinement, but missed instances in the initial masks (marked with red Xs) cannot be recovered.

to be explored if this may generalize to more complex shapes. A more robust approach may intelligently select or augment POI locations.This effect is highlighted in Figure 10. Using distance max with no perimeter buffer, where many points could be placed in over-segmented false-positive regions, many of the particles had blurred edges, including the object and its surrounding area, or even a "halo" highlighting the surrounding area but not the object, resulting in a decrease of AP@50-95 from 0.6576 to 0.2633. However, by introducing a perimeter buffer of 10%, the likelihood of false-points being placed significantly decreases, resulting in cleaner segmentations, and an AP@50-95 increase from 0.6576 to 0.8265.

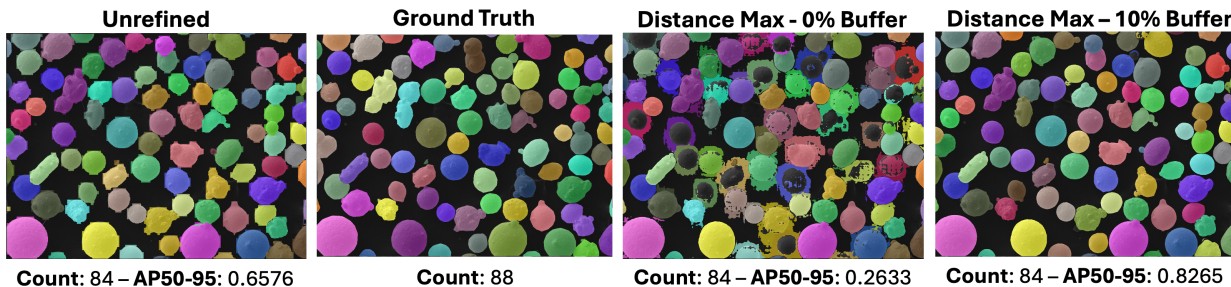

Figure 10: Example of SAM's sensitivity to false POIs, particularly for Distance-Max point placement. With no perimeter buffer, performance decreases from 0.6576 to 0.2633, but with a 10% buffer, performance increases from 0.6576 to 0.8265.

Third, SAM is most reliable when object boundaries are well defined. In images with low-contrast or gradually varying boundaries, refinement can leak across neighboring instances or into the background. As shown in Figure 11, where cells exhibit gradual border transitions, SAM blends several annotations in the bottom right corner, which did not occur for CascadePSP or SegRefiner. For this reason, SAM was not as consistently outperforming CascadePSP and SegRefiner for the additional datasets on all metrics. Despite this, SAM still improved over the unrefined masks and exceeded CascadePSP or SegRefiner on some metrics, such as AP@75.

Fourth, our work is focused on scientific images, and more specifically, microscopy images. Given SAM's domain-dependent performance, future works should evaluate the generalizability of these prompt-augmentation strategies on additional domains. In support of this, we have released our reusable code, enabling researchers to rapidly extend these results for their target domains of interest.

Lastly, although we comprehensively explored 2,688 prompt configurations across four base segmentation models (10,752 total evaluations), our search was not exhaustive. There is a possibility that alternative prompting strategies may yield further improvements.

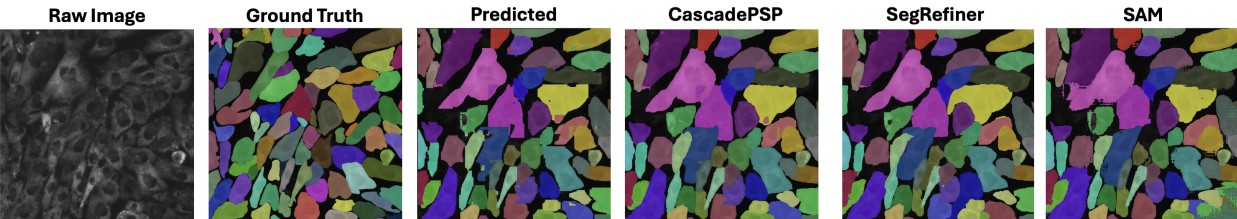

Figure 11: Qualitative example of SAM's limitations for segmenting objects with gradual boundaries, such as the cells in the lower right corner, where multiple objects are blended together.

# 6 Conclusion

Accurate object detection and segmentation are necessary for numerous tasks ranging from high-throughput object quantification to autonomous driving. Recently, there has been growing attention to model-agnostic segmentation refinement methods and post-processing detections for improved performance. In this work, we provide statistically significant evidence that SAM can perform comparably or better than state-of-the-art refinement techniques, like CascadePSP and SegRefiner, by leveraging an appropriate visual prompt configuration. However, SAM's performance depends on the prompts it is given. Recent works have conducted a preliminary analysis on the effect of prompting, but are limited in that they frequently evaluate fewer than ten prompt combinations (Cheng et al., 2023a; Hu et al., 2023; Mayladan et al., 2023). They have also rarely studied the combined effect of more than one prompt type together (e.g., points, boxes, and masks together) (Cheng et al., 2023a; Dai et al., 2023). In this work, we provide the first large-scale analysis, including 2,688 prompt combinations across four initial segmentation models, resulting in a total of 10,752 evaluations.

Through this experimentation, we uncover several key takeaways that lead to guidelines for prompt design, improving segmentation performance for scientific applications, summarized here:

- Prompts including a bounding box have an improvement of 0.320 AP@50-95 over identical prompts without a box across the 10,752 evaluations on microscopy datasets.

- Prompts including a coarse mask have a decrease in performance of -0.133 AP@50-95 over identical prompts without a mask.

- When no bounding box is provided, placing POIs along the boundary of initial segmentations is detrimental to performance, but can be mitigated by a perimeter buffer, or POI placement algorithm, such as Voronoi, that prioritizes more central regions.

- Iterative refinement was not found to yield improved performance over the highest-scoring refined mask.

These results provide a statistically significant evaluation of how prompt-augmentation can improve segmentation performance. Moving forward, this research should be extended to study other domains of interest, further verifying its generalizability. Future work can also focus on developing additional solutions for POI placement, which may mitigate the challenge of "false POIs" placed outside the ground truth.

# 7 Broader Impacts

This work poses no direct ethical or societal risks, as it operates in a data-agnostic, model-agnostic setting focused strictly on segmentation refinement. However, the underlying Segment Anything Model (SAM) is energy-intensive: its pretraining required 68 hours on 256 A100 GPUs, and each refinement pass is heavier than a single-stage segmenter. To limit inference-time carbon footprint, we recommend (1) reserving SAM-based refinement for cases where pixel-level boundary accuracy is essential, and (2) reusing SAM's large encoder embeddings across multiple objects rather than re-computing them for each mask.

## 8   Code Availability

All code and refinement results used for analysis in this study are available at: `https://github.com/sprice134/MasteringSamPrompts`

## 9   Acknowledgments

This research was supported in part by the National Science Foundation under Grant NRT-HDR-2021871, and in part by the National Science Foundation Graduate Research Fellowship Grant 2024369094. We thank Dr. Kyle Tsaknopoulos and Dr. Kiran Judd for their help with data collection and their insights into materials characterization, as well as the DAISY Research Lab and the Cote Research Lab for their feedback and support.

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
