# OpenReview forum: "Mastering SAM Prompts: A Large-Scale Empirical Study in Segmentation Refinement for Scientific Imaging"
_TMLR — Accepted by TMLR_

### Review · Reviewer_MMzu · 2025-08-28

**Summary Of Contributions:**

This paper presents an extensive empirical study on the Segment Anything Model （SAM）along with its prompt use. Specifically, 2,688 prompt configurations covering three different kinds of prompts are conducted, along with 10,752 evaluations. As a result, some valuable prompt configurations can be observed. The conclusion is valuable for future work to adapt SAM on the target domains.

**Additional Comments:**

N/A

**Audience:**

Yes

**Audience Explanation:**

The conclusion is valuable for future work to adapt SAM on the target domains.

**Claims And Evidence:**

Yes

**Claims Explanation:**

The prompt configurations and evaluations in this paper are very extensive.

**Requested Changes:**

- It is very clear that the proposed refinement framework contains many key steps or components, as shown in Fig.2. However, there is no explicit ablation study or table in this paper, that could systematically discuss and analyze the impact of each step or component.

- Besides, this paper lacks visual prediction results under these different settings with different steps.

- More segmentation results from various image domains, e.g., different medical image modalities, driving scene images, should be provided, to convince the readers with the huge working load and configurations.

- Some recent surveys that analyze SAM on multiple domains are missing, for example:
[1] Ji, Wei, et al. "Segment anything is not always perfect: An investigation of sam on different real-world applications." (2024): 617-630.
Please enrich the related work and highlight the difference over the prior works.

- For the limitation subsection Sec.5.4, it would be more convincing to also give some failure cases on each scenario.

- For the results in Table 3, a more detailed discussion should be presented, to discuss why the proposed method degrades when compared with other existing solutions.

- For the results and analysis in Sec.5.2, could the authors maybe provide some visual examples and illustrations for each subsection, so as to enhance the clarity?

- I think the authors use a low-resolution version of Fig.5. Please use a high-resolution file to compile the submission.

---

> ### Author Response · Authors · 2025-10-05
> **Author Response to Review MMzu**
>
> Thank you for your detailed review and recommendations. We have outlined the requested changes, which are color-coded blue in the manuscript.
> * Comment 1:
>     * **Review:** It is very clear that the proposed refinement framework contains many key steps or components, as shown in Fig.2. However, there is no explicit ablation study or table in this paper, that could systematically discuss and analyze the impact of each step or component.
>     * **Response:** We thank you for your suggestion. We now provide an ablation study evaluating the performance impact of each component of our recommended final prompt composition to supplement the per-component evaluations, as depicted in Table 3.
>     * **Review:** This paper lacks visual prediction results under these different settings with different steps.
>     * **Response:** As per your suggestion to show more visual displays of our results, we have added additional figures highlighting different conditions for improved visibility and interpretability, namely, Figure 6, Figure 9, Figure 10, and Figure 11.
>     * **Review:** More segmentation results from various image domains should be provided to convince the readers with the huge working load and configurations.
>     * **Response:** Thank you for your suggestion. In response to Review f9h5, we have tightened the language to narrow the scope of the proposed work and more clearly highlight this was an evaluation of scientific data, and more specifically, microscopy data. With this in mind, to test generalizability, we now also include results on ECSSD, a salient object detection dataset, highlighting a range of images including different types of objects in a diverse range of backgrounds, as shown in Table 15.
>     * **Review:** Some recent surveys that analyze SAM on multiple domains are missing, for example: [1] Ji, Wei, et al. "Segment anything is not always perfect: An investigation of SAM on different real-world applications." (2024): 617-630. Please enrich the related work and highlight the difference from the prior works.
>     * **Response:** Thank you for your comment. Based on your suggestion, we have added a new Subsection 2.3 in the Related Work section that includes this reference, as well as a broader review of published works on SAM. We discuss these prior works there. To summarize here, the key difference between our work and the recommended citation (and similar papers) is that those works primarily provide high-level surveys of _where_ SAM has been applied, noting settings where SAM performs well and where it struggles, sometimes with small illustrative examples. In contrast, our work is a systematic, large-scale experimental study focused on _how to optimize_ SAM prompting, with quantitative ablations and statistical significance testing. Here, our main contribution is the empirical analysis of prompt configurations rather than an exhaustive survey of potential domains.
>     * **Review:** For the limitation subsection Sec. 5.4, it would be more convincing to also give some failure cases on each scenario.
>     * **Response:** Based on your suggestion, we now illustrate the failure cases we had identified with example images in Figures 9, 10, and 11. These cases highlight the limitations discussed, specifically how SAM is constrained by the initial segmentation, and how false prompts can quickly deteriorate performance.
>     * **Review:** For the results in Table 3, a more detailed discussion should be presented to discuss why the proposed method degrades when compared with other existing solutions.
>     * **Response:** The reason for this degradation is that SAM looks for discrete boundaries. However, in the additional datasets, there were images of cells with a gradient boundary, increasing the difficulty of refinement for all methods, but particularly for SAM. We have expanded the discussion of this in Section 5.5, and included Figure 11 to highlight this observation.
>     * **Review:** For the results and analysis in Sec.5.2, could the authors maybe provide some visual examples and illustrations for each subsection, so as to enhance the clarity?
>     * **Response:** We have added Figure 6 in Section 5.2, which illustrates visually how a change in the number of points affects the placement of each algorithm. Additionally, we have added Figure 10, which illustrates how a change in perimeter buffer can increase or decrease performance.
>     * **Review:** I think the authors use a low-resolution version of Fig.5. Please use a high-resolution file to compile the submission.
>     * **Response:** Thank you for your close attention to detail. We have replaced Figure 5 with a higher resolution version.

---

> > ### Comment · Reviewer_MMzu · 2025-10-06
> > **Re: Author Response to Review MMzu**
> >
> > Dear Authors
> >
> > Thanks for your response letter.
> >
> > My concerns have been properly addressed.
> >
> > I recommend its acceptance.

---

### Review · Reviewer_GYZY · 2025-09-05

**Summary Of Contributions:**

Performs a large scale empirical study of the impact of prompting strategies on Segment Anything (SAM) segmentation refinement, across a wide range of prompting modalities, models, and datasets. The analysis reveals statistically significant impacts of including bounding boxes (positive) and coarse mask (negative) in the prompt.

**Audience:**

Yes

**Audience Explanation:**

SAM is used widely in the computer vision community, but has shown somewhat mixed results on scientific and microscopy datasets. By focusing its analysis on a microscopy dataset and further demonstrating generality to other microscopy datasets, this paper provides valuable insights towards extending the usefulness of SAM for scientific images.

**Broader Impact Concerns:**

No broader impact concerns

**Claims And Evidence:**

Yes

**Claims Explanation:**

The paper performs an incredibly comprehensive empirical study and rigorous statistical analysis and so backs up its claims very clearly. Its findings are concrete, prescriptive, and convincing, and likely to be of value to researchers and practitioners alike.

**Requested Changes:**

This is a very well conducted and written empirical study. The only experiment I found lacking was demonstrating whether the proposed findings also extend to future generations of SAM (eg. SAM-2). It will be valuable to evaluate whether or not that holds true, and explore potential reasons.

---

> ### Author Response · Authors · 2025-10-05
> **Author Response to Review GYZY**
>
> Thank you for your time and feedback. We have outlined the requested changes, which are color-coded violet in the manuscript.
> * Comment 1:
>     * **Review:** This is a very well conducted and written empirical study. The only experiment I found lacking was demonstrating whether the proposed findings also extend to future generations of SAM (eg. SAM-2). It will be valuable to evaluate whether or not that holds true, and explore potential reasons.
>     * **Response:** This is an interesting idea that we had not explored initially. While SAM2 was introduced as an upgrade to SAM, it was specifically designed to process videos. While it does still have single still-image capabilities, this was not the primary design motivation for SAM2. Other work in the literature had thus pointed out that SAM2 had decreased performance on single images (Pei et al. 2024). Following your suggestion,  we have conducted additional experiments comparing SAM2 to SAM performance using our optimal prompt composition on SAM2. We include our results in Section 5.4.1.

---

### Review · Reviewer_f9h5 · 2025-10-01

**Summary Of Contributions:**

This manuscript presents a large-scale empirical study on prompt design for the Segment Anything Model (SAM), specifically applied to the task of segmentation refinement. The authors systematically evaluate 2,688 distinct prompt configurations, combining points, bounding boxes, and coarse masks with various augmentation strategies across four initial segmentation models.

**Audience:**

Yes

**Audience Explanation:**

The findings are highly relevant to TMLR’s audience for several reasons. First, SAM is a widely used Foundational Model, so understanding how to effectively prompt it is of broad interest. Second, many real-world applications require high-precision segmentation, and model-agnostic refinement is a practical and active research area. Third, the scale and statistical rigor of the study address a clear gap in the literature and provide actionable insights for practitioners and researchers.

**Broader Impact Concerns:**

The paper raises no major ethical or societal concerns. The authors appropriately note the computational cost of SAM and recommend its use only when high precision is necessary. I think the work itself is methodological and domain-neutral, posing minimal risk of misuse.

**Claims And Evidence:**

Yes

**Claims Explanation:**

1.	The primary strength of this paper is its unprecedented scale and methodological rigor. Evaluating 10,752 unique model-prompt combinations provides a robust dataset from which to draw statistically significant conclusions, a clear advancement over prior ad-hoc studies.
2.	The results are distilled into clear, practical guidelines. The key takeaways are well-supported by the data and presented with statistical significance tests (paired t-tests, confidence intervals), making the conclusions highly convincing.
3.	The paper excels in its analysis of how different prompt components interact. This systemic approach is more insightful than studying components in isolation.
4.	The study is well-designed, using multiple base models, datasets, and performance metrics (AP@50, 75, 95, 50-95) to demonstrate generalizability. The comparison against established methods like CascadePSP and SegRefiner provides crucial context for the claimed performance improvement.
5.	The writing is generally clear. The prompt construction and augmentation methods are described in detail with mathematical formulations and algorithms, which enhances reproducibility.

**Requested Changes:**

1.	The most significant limitation is the focus exclusively on scientific microscopy images (powder particles, cells). While this is a valuable and challenging domain, the title and abstract suggest broader applicability. The findings, particularly the strong negative effect of coarse masks, may not hold for natural images with different object characteristics (e.g., complex shapes, occlusions). It could be better to revise the title and introduction to more accurately reflect the domain-specific nature of the study.
2.	The description of Distance Maximization and Voronoi placement algorithms (Algorithms 2 and 3) could be more intuitive. A brief high-level explanation before the pseudocode would improve readability.
3.	The chosen ranges for parameters (e.g., 1-7 points, specific scaling factors for boxes/masks) are somewhat arbitrary. A brief justification for these ranges, perhaps based on pilot studies or heuristics, would strengthen the experimental design.

---

> ### Author Response · Authors · 2025-10-05
> **Author Response to Review f9h5**
>
> Thank you for your time and thoughtful response. We have outlined the requested changes, which are color-coded orange in the manuscript.
>
> * Comment 1:
>     * **Review:** The most significant limitation is the focus exclusively on scientific microscopy images (powder particles, cells). While this is a valuable and challenging domain, the title and abstract suggest broader applicability. The findings, particularly the strong negative effect of coarse masks, may not hold for natural images with different object characteristics (e.g., complex shapes, occlusions). It could be better to revise the title and introduction to more accurately reflect the domain-specific nature of the study.
>     * **Response:** We appreciate your comment and agree. We have thus adjusted the title of our manuscript from "Mastering SAM Prompts: A Large-Scale Empirical Study in Segmentation Refinement" to "Mastering SAM Prompts: A Large-Scale Empirical Study in Segmentation Refinement for Scientific Imaging". Similarly, we have narrowed the scope of claims in both our abstract and introduction by emphasizing that our study focused on scientific images. Further, we include an additional discussion in Section 5.1.2 to indicate that further work is needed in the future to validate the generalizability of our coarse mask results on alternate domains. Lastly, in response to one of the other reviewers' suggestions to expand to other domains beyond scientific
>  images; we have conducted additional experiments on a non-scientific dataset, which have been added in Table 15.
> * Comment 2:
>     * **Review:** The description of Distance Maximization and Voronoi placement algorithms (Algorithms 2 and 3) could be more intuitive. A brief high-level explanation before the pseudocode would improve readability.
>     * **Response:** Thank you for your suggestion to increase the readability of our manuscript. We have added some intuition to both methods, including a more in-depth rationale for their design (See Section 3.3).
> * Comment 3:
>     * **Review:** The chosen ranges for parameters (e.g., 1-7 points, specific scaling factors for boxes/masks) are somewhat arbitrary. A brief justification for these ranges, perhaps based on pilot studies or heuristics, would strengthen the experimental design.
>     * **Response:** We agree that explaining our parameter ranges will be beneficial for future users of our work. We have thus included a description in Section 4.4 motivating the specific design of our experiments in general and the choice of the parameter ranges in particular. These choices of range values are supported by our experimental results. For example, as shown in Table 9, improvements began to plateau after 4 points, even decreasing going from 6 to 7 points on random placement, so we would not expect to see significant improvements by increasing $k$ beyond 6 or 7. Similarly, as shown in Table 6,  no effect was observed by eroding/dilating the coarse mask before using it as a prompt.

---

### Decision · Action_Editor_uWd1 · 2025-11-10

**Recommendation:** Accept as is

**Additional Comments:**

In this paper, the authors presented an empirical study in segmentation refinement for scientific imaging. Specifically, a large-scale study evaluating the impact of Segment Anything Model (SAM) prompt configurations was presented. Various prompt configurations with extensive evaluations were included in the study. The authors draw statistical conclusions from the experimental results, which will be of interest to the audience of TMLR.

This paper was reviewed by three expert reviewers. After a few rounds of revision and discussion between the authors and reviewers, all three reviewers recommended positively, with 2 Accept and 1 Leaning Accept. The reviewers acknowledged that their concerns were addressed by the authors' response and revision. Given the contributions this paper made and its potential interest to the audience, as well as the validated claims, the AE is happy to recommend an Accept.

**Audience:**

Yes

**Audience Explanation:**

Given the topic of this submission, there would be at least some individuals in TMLR's audience who would be interested in knowing the findings of this paper.

**Claims And Evidence:**

Yes

**Claims Explanation:**

The claims made in the submission are supported by accurate, convincing and clear evidence.